# Skill-Biased Technological Change and Gender Inequality across OECD Countries—A Simultaneous Approach

Manuel Carlos Nogueira [1,2,*] and Mara Madaleno [2]

1  ISPGAYA-Higher Polytechnic Institute of Gaya, Avenida dos Descobrimentos, 303, Santa Marinha, 4400-103 Vila Nova de Gaia, Portugal
2  GOVCOPP–Research Unit in Governance, Competitiveness and Public Policy, Department of Economics, Management, Industrial Engineering and Tourism (DEGEIT), University of Aveiro, 3810-193 Aveiro, Portugal
*  Correspondence: mnogueira@ispgaya.pt or manuel.carlos.nogueira@ua.pt

**Abstract:** Of the various approaches that, over the last few decades, have sought explanations for the constant increase in the wage gap between more and less skilled workers, the Skill-Biased Technological Change (SBTC) approach has been the most used and the one that has led to the most consistent results. The objective of this study is to assess whether the possible mobility between different types of workers, considering their experience and professional training, and this way, replacing more skilled workers in terms of education widens or reduces the wage gap between qualifications. For this purpose, we resorted to the modeling of simultaneous equations taking into account the OECD countries between 2007 and 2020, concluding that there is a strong influence of the wage gaps of the less qualified in the widening of the gaps of the more qualified and that this influence is more significant in the case of women. Education continues to promote the increase in wage differences in favor of the most qualified, as well as the SBTC approach. We also conclude that women's wage gaps are approaching the average of most workers, thus reducing wage inequality between genders.

**Keywords:** skill-biased technological change; gender wage inequality; education; simultaneous equations

## 1. Introduction

Mainly since the late 1960s, relative wages in labor markets have undergone major changes, which have caused constant increases in wage gaps in favor of more skilled workers, concerning less skilled workers, first in developed countries and then in developing countries, and this gap is called the qualification premium (Pavcnik 2017).

This increase in wage inequality has provoked researchers' interest in understanding its causes and consequences. Since the 1990s, economists have sought explanations for the fact that technological progress is no longer neutral in the distribution of earnings from work. As a result, in addition to the explanations that arise from the traditional theories of trade and the specialization of productive factors, the model by Acemoglu and Autor (2011) emerged, which, in addition to relaunching the Skill-Biased Technological Change (SBTC) approach, which in itself explains the increase in the relative demand for skilled workers, adds the functioning of labor market institutions that will influence the relative supply side of qualifications, through training policies, unions, and national or sectoral minimum wages.

On technological change, Violante (2008) and Acemoglu and Autor (2011), among others, state that the rise of technology in the workplace has increased productivity and the demand for highly qualified workers to work with this technology. In this increase in demand for highly qualified workers, there was an imbalance in the supply of qualifications,

which raised wages. Murphy and Topel (2016) state that the imbalance in the labor market between the demand for qualified workers and the respective supply has persisted over the last few decades. The SBTC is like an invisible hand in the market that regulates this wage gap in favor of more qualified workers, and as technological knowledge increases, the gap between the demand for increasingly qualified labor and less qualified and, consequently, the salary gap (Acemoglu and Autor 2011). In addition, technological advancement induces the demand for increasingly qualified workers, causing the so-called qualification premium (Acemoglu and Autor 2011). According to Violante (2008), this qualification award places technological change at the center of the debate on the distribution of income from the labor factor.

Furthermore, according to Krueger (1993) and Jorgenson (2001), in empirical terms, the technological knowledge approach (SBTC), unlike the international trade approach, is difficult to prove, given that it is a non-quantifiable variable directly. In this case, the most used proxy in the literature to measure this variable is the investments made in R&D. This variable is also referenced in the literature to measure the level of development of a country, industry, or company. Several empirical studies find positive and statistically significant coefficients for this variable (Machin and Reenen 1998; Autor et al. 1998; Violante 2008; Michaelsen 2011; Nogueira and Afonso 2018), thus proving the importance of R&D in increasing the wage gap.

Moreover, the increase in the number of qualified workers makes the market more competitive and, consequently, the companies that operate in it (Acemoglu 2002). Companies are therefore encouraged to increase their investments in technologies that use skilled workers. There is, therefore, an appetite for companies to invest in R&D activities, taking advantage of the most qualified workforce, and this investment causes an increase in their productivity. Thus, in this way, a virtuous cycle is generated, which according to Moore and Ranjan (2005), is intensified by the replacement of medium-skilled workers by more qualified ones, leading the former to unemployment, thus originating, as Michaels et al. (2014) refer, polarization in the labor market.

In the case of the US, the first authors to look into the origins of the wage gap between more qualified and less qualified workers were Katz and Murphy (1992), also the first authors to introduce the concept of SBTC, which means a bias in technological knowledge that favors more qualified workers and widens wage inequalities in their favor.

According to Katz and Murphy (1992), the increase in wage inequality between more qualified and less qualified workers seen in the US from 1963 was due to the sudden demand for workers with a university degree, which, given the constant supply in the short term, led to an imbalance in the labor market and a consequent increase in wages for qualified workers, which the authors designated as a wage premium from college.

On the other hand, Wood (1998) and Acemoglu (2002) also found that between the 1970s and 1980s, the SBTC approach was the main explanation for the increase in the gap in the US, but in other developed countries such as Japan, Korea, and France that have access to the same technology as the US, the growth of the wage gap was not as pronounced. Parnastuti et al. (2013) attribute this difference to the fact that the supply of skilled labor in some countries has increased faster than in the US. On the other hand, the SBTC approach failed to explain wage inequality in terms of race, gender, or age (Card and Lemieux 2001). However, by incorporating other variables such as unionization rates, the existence of minimum wages, or professional training, some responses emerge that complement the increase in the wage gap through the SBTC approach (Card and Lemieux 2001; Autor et al. 2008).

More recently, following theoretical modeling, Pi and Zhang (2018) found that a magnitude of technological change with a qualification bias will expand the wage gap between qualified and unskilled workers if the distributive share of labor in the skilled sector is large enough relative to the qualified sector. Even more recently, Buera et al. (2022) consider that intensive investments in skills are associated with an increase in the demand for highly qualified workers, being one of the main drivers that caused in the past, cause in

the present, and will cause in the future the increase of the labor premium qualifications, exacerbating the ability bias in the face of upward pressures on this premium.

To the best of our knowledge, this is the first time that the formation of the wage gap between more qualified and less qualified workers is addressed in the way we do in this paper. Our study refers to OECD countries from 2007 to 2020 and incorporates a wide range of control variables in addition to the SBTC proxy variable. We estimated four models, two of them with the wage gap between university and high school graduates and the wage gap between workers with high school graduates and below high school graduates for most workers in OECD countries. The other two models are similar to the first, except that they refer to wage gaps for female workers. Another novelty is that modeling through simultaneous equations is used for the first time. We decided on this approach as we suspect there may be simultaneous relationships between these wage gaps.

Some workers with average qualifications may occupy positions of workers with higher qualifications through the accumulation of experience, training, or professional education. In this way, as there can be mobility between classes of workers taking into account their qualifications, the use of simultaneous equations incorporating wage rates as independent variables and wage gaps as independent variables are justified.

In our opinion, the choice proved to be the right one since strong evidence of positive influences of the wage gap of less qualified workers was found on the wage gap of more qualified workers. Furthermore, the wage gap between medium-skilled and less qualified workers promotes an increase in the wage gap concerning the most qualified, showing that despite obtaining professional training, thus raising salary, the market rewards the qualifications obtained in tertiary education.

In addition to these conclusions, other important evidence was found. Expenses incurred with investments in education cause an increase in wage gaps in favor of more qualified workers, and economic growth causes a reduction in this gap. We found that unionization only has effects in reducing the wage gap of less qualified workers, that globalization does not significantly impact the wage gap, and that environmental variables either do not interfere with wage gaps or have a reduced impact.

Another invading factor in our study is that, as far as we know, this is the first time that when workers are divided into three types of qualifications, it is done separately for most workers and female workers. This allows us to verify which variables, as well as their intensity, better justify the formation of the wage gap for most workers and female workers since women in private companies generally enjoy lower wages than men, even with equal qualifications.

Traditionally in the economic growth literature, technological progress is considered a factor that favors all workers, increasing their productivity, and is seen as the main long-term determinant of income levels. However, what has been seen is that technological change tends to favor qualified workers via endogenous factors such as the demand for and supply of qualifications, placing the distribution of workers' general income at the center of the debate (Violante 2008). Autor (2019) considers this reality a lasting paradox since, for more than four decades in industrialized economies, the real wages of less qualified workers have shown sustained declines, while workers with higher qualifications have seen their real wages increase. According to Acemoglu and Restrepo (2021), the most popular explanation for this reality is based on the Skill-Biased Technological Change (SBTC) approach.

After these introductory remarks, the paper analyzes the literature (Section 2). Then, Section 3 presents data, variables, statistics, and correlations. Next, Section 4 presents the empirical analysis, Section 5 presents the discussion of the results, and finally, Section 6 concludes the paper and presents some policy implications.

## 2. Literature Review

The formulation of wage inequality between more and less skilled workers has had a vibrant debate in the economic literature. Not exclusively, but over the last few decades,

two main approaches have emerged that seek to justify the formation of these wage gaps. The first approach arose through the liberalization of international trade and originated in the insights of the Stolper–Samuelson Theorem (Borjas et al. 1997). However, more recently and largely due to studies by Acemoglu (2002), there has been an increase in the Skill-Biased Technological Change (SBTC) approach, which ended up overlapping the international trade approach.

### 2.1. Skill-Biased Technological Change and the Wage Gap

According to Lemieux (2007) and Grossman and Helpman (2018), the increasing emergence of new technologies enhances the relative demand for more qualified labor, which induces a complementarity between highly productive capital goods, such as ICT, and more skilled workers. The relative increase in demand for this type of worker exceeds their relative supply, thus increasing the skills premium. Çaliskan (2015) states that the differences in economic growth between countries are explained based on the technologies each uses. These play a key role in reducing costs and increasing productivity, reinforcing the importance of studying the role of technological wage level. However, not always, and in all geographies, as well as in all time spaces, there is an increase in wage inequality between skilled and unskilled workers. For example, Messina and Silva (2019), for a large group of 16 countries in Latin America, concluded that between 1995 and 2002, wage inequality increased in most countries. However, from 2002 until 2015, on average, this inequality was reduced by 26%, mainly due to wage reductions for higher education graduates (due to a rapid expansion in educational attainment, which increased the supply of skilled workers relative to demand needs); and that wage increases in the lower percentiles of the distribution of each country's wage rate (due to relative increases in minimum wages) (Messina and Silva 2019).

In other geographies, the conclusions are different. For example, according to Hutter and Weber (2022) and the case of Germany, which analyzed the effects of SBTC on the labor market, for the period between 1975 and 2014, SBTC is a source of increased productivity and wages, causing increases in wage inequalities in favor of more skilled workers. Moreover, for the same authors, the wage gap has increased mainly since the 1990s, still showing the reduced hours worked by the most qualified due to their higher productivity.

These things considered, recent literature has added other factors and other justifications for the increase in the wage gap, which continues to be verified. For example, the level of suppression or replacement of routine tasks that were performed manually by unskilled workers and which are now performed using automation increases the wages of skilled workers who perform cognitive tasks and reduces or extinguishes the jobs of unskilled workers, further widening wage gaps (Acemoglu and Restrepo 2018, 2021). In addition, explanations from the side of entrepreneurship (Naudé and Nogler 2018) or offshoring tasks in global value chains (Wang et al. 2021) allow for widening wage inequality. Tyrowicz and Smyk (2019) found that between 1980 and 2010, wage inequality among workers was lower in transition economies. Still, when they come to be considered developed economies, the wage gap widens in favor of skilled workers.

Broecke et al. (2015) state that it is not only in the US that wage inequality is on the rise but also in OECD countries. For these authors, workers in the 90th percentile earn, on average, 3.4 times more than workers in the 10th percentile (considering all OECD countries). Moreover, based on OECD countries, Nogueira and Afonso (2018) conclude that for clusters composed of countries with the highest GDP per capita, the SBTC approach promotes the widening of the wage gap between university graduates (skilled workers) and high school graduates (unskilled workers). Still, on the proliferation of job polarization, Broecke et al. (2015) state that permanent technological change causes an increase in the demand for increasingly qualified workers. Supply cannot immediately keep up with demand, and this causes the wage gap to widen. Furthermore, as routine tasks are becoming increasingly automated, the demand for mid-level qualifications has declined, causing a polarization of employment and exacerbating the trend of wage inequality. For



Broecke et al. (2015), the costs associated with this growing inequality relate to reduced social mobility, community problems, reduced social cohesion, and increased crime.

### 2.2. Wage Inequality Driving Factors

Other authors, such as Western and Rosenfeld (2011), emphasize that institutional factors drive inequality in addition to the SBTC approach. The decline in unionization, the fall in the real value of the minimum wage, the spread of non-standard employment practices, the rise of financialization, the outsourcing of work, and the corresponding decline in internal labor markets and globalization also contribute to the wage gap. A real fall in the values of the lowest wages causes an increase in inequality, not by increasing the value of the highest wages but by reducing the lowest wages in real terms. Thus, technological and institutional factors are not mutually exclusive but complement each other.

There is a broad consensus that the decline of organized work on a union basis has increased wage inequality. Autor et al. (2008) point out that institutions that set minimum wages, the high existence of trade unions, and professional training for unskilled workers, among other factors, protect unskilled workers by artificially sustaining the wages of these workers. Moreover, Card et al. (2013) attributed a major contribution to the increase in wage inequality to the decline in unionization. Still, they included other factors such as the decrease in real values of minimum wages, the increase in heterogeneity in the workplace, the rise of international trade, and the very changes in the institutions that regulate the labor market. In addition, the SBTC approach also contributes to wage dispersion among workers taking into account their education (Card et al. 2013). Still on the role played by the large reduction in the number of unionized workers, Biewen and Seckler (2019), and in the case of German companies between 1995 and 2010, attribute this decrease as the main reason for the increase in wage inequality. Furthermore, the increase in schooling, changing tasks, the internationalization of companies, and their heterogeneity contribute to the increase in this gap.

Reinforcing the importance of the drop in union membership in exacerbating the wage gap, Western and Rosenfeld (2011) state that for unionized workers, unions directly reduce the wage gap by fighting for wage increases and indirectly increase the wages of workers, non-union members, down to the union level to avoid unionization. In this way, they set a pattern for wage increases across the industry and the emergence of legislation favorable to low-income and less-skilled workers. These authors conclude that the decline in unionization explains about a third of the increase in wage inequality in men and about a fifth in the case of women. In addition, for the US, Kristal and Cohen (2017) argue that the decline of unions and the fall in the real value of the minimum wage explain about half of the increase in wage inequality, while the SBTC approach explains about a quarter of this increase. These findings explain that a large part of the increase in inequality in the US is driven by the low power of unskilled workers rather than by market forces.

Among economists, it is practically unanimous that technical and consequent technological progress determine economic growth via innovation, knowledge, and productivity (Korres 2008). Previously, Gomulka (1990) considered that even considering the institutional and cultural characteristics of different countries and in the light of other economic theories, there is a positive relationship between technological progress and economic growth in both the short and long term. Solow (1957), using an aggregate production function, found that 87% of the variation in economic growth in the US would be due to technological progress (a fact known as the Solow residual).

### 2.3. Other Factors Able to Explain Wage Inequality

The conciliation between economic growth and environmental preservation that guarantees sustainability and the preservation of resources has not always been peaceful. Moreover, it has even been considered an authentic trade-off, since economic growth, by stimulating production and consumption activities, generates additional polluting emissions that degrade the environment and consume more natural resources (Marsiglio

and Priveleggi 2021). When humanity became aware that the planet's resources are scarce and that pollution causes climate change, it was necessary to start finding alternatives for socioeconomic development with environmental protection. Some authors have defended introducing high carbon taxes for the most polluting industries. Still, for the critics of this literature that follows the neoclassical marginal analysis, the regulation of environmental protection will encourage investment in production technologies that are less harmful to the environment. Still, it will divert productive investment undermining economic growth (Tang et al. 2019). In the same sense, authors such as Greenstone et al. (2012) empirically verified that environmental regulation negatively affects the total factor productivity of companies and, consequently, economic growth. Calculating on a large scale the first estimates of the economic costs of environmental regulations, these authors believe that it will generate unemployment and wage reductions. In addition, environmental regulations can be considered "job killers" due to companies' loss of competitiveness (Greenstone et al. 2012).

Faced with the growing debate about the harmful effects of environmental protection on the economy, Porter (1991) argues that environmental protection regulations will increase productivity as companies rationalize their operations, triggering innovation. Complementing the opinion of Porter (1991), authors such as Özokcu and Özdemir (2017) and Bashir et al. (2021) argue that the environmental costs caused by economic growth occur only in the short term (where there is a U-shaped relationship between degradation and economic growth), and in the long term (after overcoming the transition phase) the economic growth and technology itself solve environmental problems. Marsiglio and Priveleggi (2021) also argue that technological progress can allow sustained economic growth in the long term, finding a balance between economics and environmental objectives. In a recent and extensive empirical study of all European Union countries, Nogueira and Madaleno (2021) found that it is possible to obtain economic growth without neglecting sustainability and considering environmental concerns, contradicting the opinion that there is a trade-off between these two realities.

Although very incipient, the first scientific studies are beginning to appear, seeking to relate environmental concerns and sustainability with workers' wages. One of these first studies was elaborated by Krueger et al. (2021), which states that there is growing evidence that workers are increasingly concerned with environmental sustainability and are willing to accept lower wages (about 10% less) to work in companies that operate in sectors that care about environmental sustainability, the environment, as well as in these sectors the employee retention rate is higher than in others. These authors also conclude that the more qualified the workers are, the greater their willingness to accept lower wages. These authors called this difference the "Sustainability Wage Gap". In this way, the decrease in wage costs could be channelled into investments in less polluting technologies and maintain profitability. In the same sense, Bunderson and Thakor (2020) and Schneider et al. (2020) found that only the most qualified workers are willing to give up part of their wages to work in more sustainable jobs and companies, with these companies being able to attract and retain more talented and more qualified workers.

For the case of developing countries, Ee et al. (2018) consider that the launch of high taxes on pollution may, in addition to reducing pollutant emissions, reduce the wage gap between skilled and unskilled workers, especially in the long term. Still, in the short term, wage inequality may be reduced and increased by using more qualified labor to implement measures to combat pollution.

The Environment Performance Index (EPI) ranks through quantitative metrics the performance of countries over the years on environmental issues that are considered in two dimensions of high priority: protection of human health and protection of ecosystems. The construction of the index is in line with the United Nations Sustainable Development Goals to be achieved by 2030. According to Hsu (2016) and Wendling et al. (2018), there is a positive and significant relationship between the value of the EPI index and economic growth, with the financial resources of the countries with the best score being used to protect

human health and the environment. These authors also find that the first countries in the EPI ranking are those with a higher GDP per capita. In addition, Nogueira and Madaleno (2021) and the countries of the EU found strong empirical evidence of the link between EPI value and economic growth in these countries. Thus, showing that environmental and sustainability concerns do not conflict with economic growth, seeking not to deteriorate the planet further and safeguarding future generations.

According to Constantini and Monni (2008), the first Human Development Reports did not explicitly consider the role of the environment in people's choices. Still, in editions after 2000, concerns about the environment and sustainable development were introduced in their calculation. Currently, environmental quality and sustainable development are considered determinants of well-being. The United Nations Millennium Development Goals also reinforce the full integration of human development and the environment as mutually reinforcing goals.

Several empirical and theoretical studies consider this variable regarding the expected impact of spending on education on wage inequality between different countries and levels of qualifications. The vast majority of these studies find that increases in education spending lead to increases in wage gaps (Benabou 2000; Muinelo-Gallo and Roca-Sagalés 2011). Antonczyk et al. (2018), using the case of the US and Germany, found evidence of polarization in the labor market, with education being the main hypothesis to explain the increase in wage inequality, which they call a premium for skills. Employment polarization consists of job growth at the top and bottom of the income distribution, with a consequent decrease at intermediate levels (Michaels et al. 2014). Workers who perform routine tasks are replaced by robotics and automation, which have made strong advances due to technological improvements and the fall in the price of computational capital (Acemoglu and Restrepo 2021).

Moreover, for OECD countries, but in another time frame, Nogueira and Afonso (2018) found that among the variables considered, spending on education is the variable that most impacts the wage gap between skilled and unskilled workers, showing the particularity of increasing the hiatus sharply. Furthermore, the more a country invests in education, the higher the qualifications of the students, who will receive higher wages when they leave for the job market than unskilled students, widening the wage gap between them (Nogueira and Afonso (2018). Recently Jacobs and Thuemmel (2022), when verifying, among other variables, the effects caused by state subsidies to education, found that these entail greater distributional losses (which widens wage gaps), in addition to indirectly causing greater investments in education, as the SBTC gains importance and develops.

By using some variables inherent to globalization processes such as immigration, trade, and FDI, Jestl et al. (2022) conclude that for the EU, immigration contributes to the increase in wage inequality in the center and at the top of the wage distribution. Regarding trade and FDI, the increase in inequality occurs more intensely in the older EU countries at the center and top of the wage distribution and in the new countries at the center and bottom of the wage distribution. These conclusions seem to want to show that immigrants are willing to accept lower wages than natives for all qualifications and that trade and FDI in older EU countries increase wage inequality in more qualified ones and newer countries in less qualified ones. Meschi et al. (2016) also believe that globalization increases the gap between employment and the wage level between skilled and unskilled workers. Moore and Ranjan (2005) found that, for OECD countries, globalization increased the relative price of skills-intensive goods, with technological progress (SBTC) increasing the relative marginal product of skilled workers in world production. These two combined effects contribute to the wage inequality between skilled and unskilled workers, causing increased demand for skilled workers and unemployment of unskilled workers.

O'Rourke (2001) mentioned that factors such as GDP per capita affect wage inequality between countries. Kuznets (1955) previously suggested that increases in GDP per capita should be associated with reductions in wage inequality, given that economic development will lead people with greater purchasing power to seek to invest in education and acquire

more qualifications. This variable, seen in isolation, will increase the supply of qualified workers, thus reducing their relative wages and wage inequality for unskilled workers. In the same sense, Nogueira and Afonso (2018) and also for OECD countries empirically verified that GDP per capita reduces wage inequality. The introduction of this variable in our study is also related to the fact that the OECD countries present different economic realities. More reliable and robust results will be obtained with the division of GDP by the number of inhabitants.

Wage disparities between men and women have attracted the attention of numerous studies. Although there is still a visible wage inequality between men and women, Shen (2014) finds that in the US, this inequality has been reduced in recent decades by the fact that women are acquiring a greater number of qualifications, competing with men for higher wages. These things considered, as the SBTC multiplied the economic returns for qualifications, the increasing participation of women in positions of greater responsibility, with corresponding higher salaries, was one of the causes for the decrease in wage inequality. However, this inequality has decreased for all qualifications (Shen 2014). Another factor pointed out by the author is the increase in women's economic independence, with their wider access to the labor market. In the same sense, Moore (2018) states that as a sign of progress toward gender equality, the world has witnessed a convergence between men's and women's salaries, largely due to changes in women's occupational careers, which amount to better-paid positions because of the greater responsibility that is asked of them.

More recently, Kovalenko and Töpfer (2021) refer that demand and labor supply shocks, in general terms, affect the wage gap only in the short and medium term, with increases in female labor supply increasing the gap in the short term and increases in male labor supply do not affect the wage gap. Still, for these authors, after the year 2000, the wage gap between men and women was reduced as a response to a massive entry of women into the labor market and due to the shocks of technological advances and computerization, and information technology innovation, in addition to having promoted the increase in GDP, reducing the wage gap. However, Kovalenko and Töpfer (2021) also state that the most recent advances in artificial intelligence and robotization are not contributing to wage approximation, a reality that they attribute to these technological advances being concentrated in sectors and industries dominated by men.

### 3. Data, Variables, Statistics, and Correlations

The sample we use in the empirical analysis covers 34 OECD countries and the periods from 2007 to 2020. However, countries like Iceland, Costa Rica, Lithuania, and Colombia are excluded from this sample due to the lack of data on some variables considered in the estimated models. In addition, statistical information is not uniformly given for some countries. Considering these statistical limitations, the empirical analysis uses a data estimation approach with 396 observations for all workers and 333 for women instead of the 476 observations if there were no missing data.

Table 1 explains the variables used in the empirical analysis, units of measurement, and the data source. Table 2 presents the main descriptive statistics and the correlations, and Table 3 presents the average variables for each OECD country.

To derive accurate results from the empirical analysis, we also considered the problem of multicollinearity. When applied to our variables, Pearson's correlation test (Table 2) showed no multicollinearity between the variables considered. Therefore, according to Masanipour and Thompson (2020), we used the value of 0.80 as a limit, as posited by some renowned econometricians, although there is no absolute consensus on this value.

**Table 1.** Variable definition and data source.

| Variable | Definition | Unit | Source |
|---|---|---|---|
| $WGH_{i,t}/WGM_{i,t}$ | Wage gap between university graduates and high school graduates in country i and year t, in real terms | Index | OECD Education at a Glance—Kovalenko and Töpfer (2021); Acemoglu and Restrepo (2018, 2021) |
| $WGM_{i,t}/WGL_{i,t}$ | Wage gap between high school graduates and below high school graduates in country i and year t, in real terms. | Index | OECD Education at a Glance—Kovalenko and Töpfer (2021); Acemoglu and Restrepo (2018, 2021) |
| $WGWH_{i,t}/WGWM_{i,t}$ | Wage gap between women university graduates and high school graduates in country i and year t, in real terms, as a percentage of men's earnings. | Index | OECD Education at a Glance—Kovalenko and Töpfer (2021); Acemoglu and Restrepo (2018, 2021) |
| $WGWM_{i,t}/WGWL_{i,t}$ | Wage gap between women high school graduates and below high school in country i and year t, in real terms, as a percentage of men's earnings. | Index | OECD Education at a Glance—Kovalenko and Töpfer (2021); Acemoglu and Restrepo (2018, 2021) |
| $SBTC_{i,t}$ | Research and Development spending as a percentage of GDP in country i and year t | Percentage | OECD—Acemoglu and Restrepo (2018, 2021); Kristal and Cohen (2017) |
| $Union_{i,t}$ | Share of unionized workers in country i and year t | Percentage | OECD—Kristal and Cohen (2017) |
| $EPI_{i,t}$ | Environmental Performance Index, in the country i and year t | Index | Environmental Law and Policy—Hsu (2016); Wendling et al. (2018) |
| $Educ.Expend_{i,t}$ | Education expenditure as a percentage of GDP in country i and year t | Percentage | OECD Education at a Glance—Nogueira and Afonso (2018) |
| $CO_2$ | $CO_2$ emissions per capita in country i and year t | Tons | World Bank—Nogueira and Madaleno (2021) |
| $KOF_{i,t}$ | Globalization Economic Index in country i and year t | Index | KOF Swiss Economic Institute |
| $GDP\ pc_{i,t}$ | Gross domestic product per capita in country i and year t, US dollar constant prices, 2015 PPPs | Value in dollars | OECD World Bank—Nogueira and Afonso (2018) |

Source: Authors' elaboration.

**Table 2.** Main descriptive statistics and correlations.

| | WGH | WGM | WGL | WGWH | WGWM | WGWL | SBTC | Union | EPI | Educ. Expend. | CO2 | KOF | GDPpc | Average | Standard Deviation | Max | Min |
|---|---|---|---|---|---|---|---|---|---|---|---|---|---|---|---|---|---|
| WGH | - | 0.06 | −0.48 | −0.13 | 0.11 | 0.02 | −0.36 | −0.41 | −0.24 | −0.29 | −0.23 | −0.25 | −0.40 | 154.63 | 23.361 | 260 | 115 |
| WGM | | - | −0.18 | 0.09 | 0.06 | 0.10 | 0.18 | 0.16 | 0.21 | 0.04 | 0.11 | 0.18 | 0.29 | 107.72 | 12.360 | 146 | 61 |
| WGL | | | - | 0.11 | 0.03 | 0.06 | 0.18 | 0.37 | 0.23 | 0.12 | 0.11 | 0.33 | 0.15 | 78.221 | 8.1625 | 101 | 54 |
| WGWH | | | | - | 0.38 | 0.23 | −0.03 | 0.23 | 0.16 | −0.02 | −0.06 | 0.07 | 0.14 | 75.525 | 7.1177 | 148 | 61 |
| WGWM | | | | | - | 0.64 | 0.08 | 0.25 | 0.08 | −0.21 | −0.32 | 0.26 | 0.13 | 77.080 | 6.6743 | 98 | 54 |
| WGWL | | | | | | - | 0.13 | 0.44 | 0.19 | −0.07 | −0.15 | 0.40 | 0.38 | 76.154 | 6.6814 | 92 | 49 |
| SBTC | | | | | | | - | 0.41 | 0.18 | 0.29 | 0.18 | 0.35 | 0.36 | 1.9327 | 1.0352 | 4.93 | 0.28 |
| Union | | | | | | | | - | 0.39 | 0.35 | 0.06 | 0.47 | 0.46 | 24.813 | 17.418 | 72.5 | 4.53 |
| EPI | | | | | | | | | - | 0.22 | 0.22 | 0.44 | 0.41 | 79.770 | 8.3420 | 90.8 | 42.6 |
| Educ. Expend. | | | | | | | | | | - | 0.05 | −0.13 | 0.07 | 5.4694 | 1.0424 | 8.42 | 3.25 |
| CO2 | | | | | | | | | | | - | 0.13 | 0.45 | 8.6885 | 4.0938 | 23.8 | 2.77 |
| KOF | | | | | | | | | | | | - | 0.54 | 82.021 | 5.8417 | 90.9 | 61.8 |
| GDPpc | | | | | | | | | | | | | - | 38,045 | 23,153 | 116,597 | 8002 |

Source: Authors' elaboration.

**Table 3.** Average of variables for each OECD country (2007–2020). Source: Authors' elaboration.

| Country | WGH | WGM | WGL | WGWH | WGWM | WGWL | SBTC (%) | Union (%) | EPI | Educ. Exp. | CO$_2$ | KOF | GDPpc |
|---|---|---|---|---|---|---|---|---|---|---|---|---|---|
| Australia | 132.85 | 97.35 | 83.5 | 77.72 | 75.81 | 80.01 | 2.01 | 16.26 | 83.78 | 5.67 | 17.42 | 80.54 | 55,856 |
| Austrium | 153.57 | 118.28 | 69.14 | 73.54 | 79.54 | 76.81 | 2.93 | 28.09 | 83.07 | 5.28 | 7.92 | 86.94 | 44,460 |
| Belgium | 133.71 | 99.35 | 88.78 | 81.36 | 82.63 | 80.82 | 2.54 | 52.46 | 77.80 | 6.11 | 9.31 | 89.51 | 40,622 |
| Canada | 140.42 | 113.71 | 81.71 | 72.63 | 70.54 | 66.81 | 1.75 | 26.76 | 81.22 | 6.21 | 16.08 | 82.87 | 42,771 |
| Chile | 246.51 | - | 67.25 | 66.25 | 72.00 | 78.00 | 0.36 | 14.68 | 72.12 | 6.28 | 4.41 | 76.33 | 12,826 |
| Czech Republic | 175.92 | - | 72.57 | 71.63 | 79.83 | 79.91 | 1.67 | 13.92 | 79.53 | 4.32 | 10.37 | 83.09 | 17,674 |
| Denmark | 127.14 | 102.57 | 82.42 | 77.01 | 80.54 | 81.91 | 2.93 | 68.20 | 86.55 | 7.09 | 7.19 | 87.72 | 53,587 |
| Estonia | 132.63 | 89.66 | 89.91 | 70.27 | 61.54 | 61.18 | 1.53 | 6.12 | 81.01 | 5.21 | 12.92 | 80.93 | 17,320 |
| Finland | 143.85 | 119.14 | 95.35 | 77.54 | 78.18 | 79.72 | 3.17 | 67.22 | 87.25 | 5.91 | 9.43 | 86.58 | 44,329 |
| France | 149.50 | 89.66 | 83.42 | 74.45 | 80.18 | 74.63 | 2.19 | 10.78 | 85.27 | 5.70 | 5.42 | 86.50 | 36,620 |
| Germany | 164.28 | 112.01 | 82.78 | 73.90 | 82.27 | 76.63 | 2.88 | 17.93 | 81.68 | 4.65 | 9.67 | 87.37 | 40,276 |
| Greece | 146.91 | 102.09 | 75.27 | 74.72 | 78.54 | 68.82 | 0.88 | 21.72 | 79.85 | 3.71 | 7.85 | 80.32 | 19,654 |
| Hungary | 204.01 | 109.92 | 74.42 | 72.91 | 87.72 | 83.18 | 1.24 | 11.06 | 76.38 | 4.50 | 5.03 | 84.27 | 12,575 |
| Ireland | 167.42 | 96.92 | 85.71 | 75.36 | 77.15 | 80.45 | 1.37 | 27.97 | 84.13 | 4.94 | 8.61 | 85.46 | 56,989 |
| Israel | 154.02 | 111.87 | 76.28 | 69.90 | 75.27 | 72.72 | 4.39 | 26.03 | 75.98 | 6.51 | 8.25 | 76.82 | 35,040 |
| Italy | 147.85 | - | 78.14 | 72.90 | 76.72 | 77.45 | 1.30 | 34.05 | 80.62 | 4.35 | 6.49 | 81.51 | 34,981 |
| Japan | 150.27 | - | 78.72 | - | - | - | 3.21 | 17.82 | 78.85 | 4.58 | 9.46 | 75.07 | 40,898 |
| Korea | 143.21 | - | 70.85 | 67.36 | 65.18 | 66.72 | 3.78 | 10.29 | 69.02 | 6.66 | 12.17 | 75.82 | 27,218 |
| Latvia | 145.20 | 98.40 | 88.80 | 77.40 | 71.80 | 69.60 | 0.59 | 13.49 | 78.76 | 4.42 | 4.72 | 75.03 | 14,981 |
| Luxembourg | 153.28 | 125.12 | 71.71 | 79.45 | 79.45 | 81.72 | 1.35 | 34.39 | 84.66 | 3.73 | 19.44 | 85.48 | 110,257 |
| Mexico | 192.33 | 120.35 | 62.16 | 69.16 | 77.40 | 72.66 | 0.40 | 13.90 | 67.92 | 5.59 | 3.93 | 67.07 | 9618 |
| Netherlands | 152.57 | 114.35 | 83.35 | 77.36 | 81.27 | 81.09 | 1.97 | 18.29 | 79.25 | 5.55 | 9.73 | 89.07 | 51,446 |
| New Zealand | 127.71 | 110.07 | 83.64 | 77.45 | 77.27 | 79.27 | 1.24 | 19.42 | 84.02 | 6.64 | 8.59 | 76.71 | 38,626 |
| Norway | 126.38 | 114.85 | 79.07 | 75.36 | 77.37 | 80.82 | 1.82 | 50.03 | 84.42 | 6.87 | 9.75 | 84.81 | 85,543 |
| Poland | 167.50 | 104.57 | 83.28 | 76.81 | 77.45 | 71.72 | 0.87 | 15.53 | 68.29 | 5.18 | 8.51 | 78.77 | 12,205 |
| Portugal | 166.35 | 101.14 | 70.14 | 73.54 | 74.00 | 71.74 | 1.37 | 17.60 | 74.39 | 5.53 | 4.98 | 82.39 | 19,728 |
| Slovak Repubic | 171.72 | 131.47 | 68.18 | 70.27 | 73.91 | 73.36 | 0.75 | 14.05 | 79.11 | 4.12 | 6.67 | 81.48 | 17,781 |
| Slovenia | 181.71 | - | 76.85 | 86.09 | 86.18 | 84.36 | 2.07 | 30.69 | 81.02 | 5.03 | 7.38 | 79.32 | 24,177 |
| Spain | 142.64 | 109.0 | 79.35 | 84.18 | 76.58 | 76.08 | 1.27 | 15.85 | 84.29 | 4.71 | 5.97 | 83.59 | 29,731 |
| Sweden | 123.78 | 114.85 | 83.28 | 81.18 | 81.81 | 85.03 | 3.28 | 67.40 | 86.32 | 6.02 | 4.57 | 88.78 | 54,692 |
| Switzerland | 153.50 | 109.12 | 76.35 | 78.66 | 83.83 | 78.83 | 3.11 | 16.36 | 85.64 | 5.19 | 4.98 | 89.34 | 82,481 |
| Turkey | 160.92 | - | 69.57 | 83.57 | 80.43 | 69.00 | 0.86 | 7.75 | 59.38 | 4.60 | 4.71 | 68.36 | 10,583 |
| United Kingdom | 154.53 | - | 71.21 | 76.63 | 72.36 | 74.90 | 1.65 | 25.43 | 85.66 | 6.14 | 6.92 | 88.58 | 43,096 |
| United States | 174.46 | 108.5 | 67.64 | 69.90 | 71.00 | 70.72 | 2.82 | 10.84 | 79.62 | 6.72 | 17.22 | 81.13 | 54,888 |

Considering the weighted average salary of all workers with secondary education with an index of 100, we see in Table 3 that the average of all workers with higher education presents an index of 154.63, while for women only, this value is 75.525. The weighted average wage index for workers with high school graduates is 107.72, and for women, it is 77.08. Finally, the average salary index for workers below high school graduates is 78.22, and for women, it is 76.154. Given these wage index values, we can see that the wage disparity between all workers and women has a greater impact on higher education workers. In the case of having few qualifications, the wage disparity is reduced, which is not unrelated to the existence in many countries of minimum wages that do not discriminate against men and women. We can also see in Table 3 that the standard deviation is higher for workers with a higher education course because the maximum index is 260 (Chile) and the minimum is 115 (New Zealand).

On average, the country that invests the most in R&D as a percentage of GDP is Israel, and the one that invests the least is Mexico. As regards the percentage of unionized workers, the average is highest in Denmark and lowest in Estonia. On the other hand, the Environmental Performance Index (EPI) reaches the highest average value in Finland and the lowest in Mexico. The country that invests the most in average terms and as a percentage of GDP in education is Norway, and the one that invests the least is Greece. Average per capita $CO_2$ emissions are highest in Luxembourg and lowest in Sweden. The average of the Globalization Index (KOF) is highest in Belgium, with Latvia being the country with the lowest value. Finally, in Table 3, we see the large discrepancy between the maximum and minimum GDPpc values, which occur in Luxembourg and Mexico, respectively.

## 4. Empirical Analysis, Model Specification, and Estimation Methods

As previously mentioned, we intend to verify the influence of seven variables in the formulation of the wage gap between workers in OECD countries who have university graduates and those who have only high school graduates and those who have high school graduate qualifications and those who have qualifications at the level of below high school graduates. In addition, we also intend to verify for women only (as a percentage of men's salary index) the same effects of these seven variables shown in Table 1.

As there is evidence of simultaneity relationships between university graduates and high school graduates and between these and below high school graduates, the econometric estimates were therefore carried out using a system of simultaneous equations, with the structural form of the equations being as follows:

$$\text{LnWGH}_{i,t}/\text{LnWGM}_{i,t} = \alpha_i + \beta_1 \text{LnWGM}_{i,t}/\text{LnWGL}_{i,t} + \beta_2 \text{LnSBTC}_{i,t} + \beta_3 \text{LnUnion}_{i,t} + \beta_4 \text{LnEPI}_{i,t} + \beta_5 \text{LnEduc.expend.}_{i,t} + \beta_6 \text{LnCO2}_{i,t} + \beta_7 \text{LnGDPpc}_{i,t} + \mu_{i,t} \tag{1}$$

$$\text{LnWGM}_{i,t}/\text{LnWGL}_{i,t} = \alpha_i + \sigma_1 \text{LnSBTC}_{i,t} + \sigma_2 \text{LnUnion}_{i,t} + \sigma_3 \text{LnEPI}_{i,t} + \sigma_4 \text{LnEduc.Expend.}_{i,t} + \sigma_5 \text{LnCO2}_{i,t} + \sigma_6 \text{LnKOF}_{i,t} + \sigma_7 \text{LnGDPpc}_{i,t} + \mu_{i,t} \tag{2}$$

$$\text{LnWGWH}_{i,t}/\text{LnWGWM}_{i,t} = \alpha_i + \beta_1 \text{LnWGWM}_{i,t}/\text{LnWGWL}_{i,t} + \beta_2 \text{LnSBTC}_{i,t} + \beta_3 \text{LnUnion}_{i,t} + \beta_4 \text{LnEPI}_{i,t} + \beta_5 \text{LnEduc.Expend.}_{i,t} + \beta_6 \text{LnCO2}_{i,t} + + \beta_7 \text{LnGDPpc}_{i,t} + \mu_{i,t} \tag{3}$$

$$\text{LnWGWM}_{i,t}/\text{LnWGWL}_{i,t} = \alpha_i + \sigma_1 \text{LnSBTC}_{i,t} + \sigma_2 \text{LnUnion}_{i,t} + \sigma_3 \text{LnEPI}_{i,t} + \sigma_4 \text{LnEduc.Espend.}_{i,t} + \sigma_5 \text{LnCO2}_{i,t} + \sigma_6 \text{LnKOF}_{i,t} + \sigma_7 \text{LnGDPpc}_{i,t} + \mu_{i,t} \tag{4}$$

Equation (1) regresses the salary gap between university and high school graduates. It includes, in addition to six of the seven independent variables identified in Table 1, the salary gap between high school graduates and below-high-school graduates. Equation (2), in turn, regresses the wage gap between high school graduates and below high school graduates and also includes six of the seven independent variables in Table 1. Equations (3) and (4), in turn, present the same variables as Equations (1) and (2) but refer only to female workers as a percentage of male workers' earnings.

The approach through simultaneous equations makes it possible to analyze the individual behavior of each equation and the possible relationships between equations and variables in a given period. In this way, it becomes possible to increase the accuracy of the model's estimates, using additional information provided by the interrelationships, providing a more reliable measure.

To carry out the identification of the structural equations of the system of simultaneous equations, the condition of order was considered since, according to Gujarati and Porter (2008), in practical terms, this condition is generally adequate to guarantee the identifiability in case the number of equations is only two.

As shown in Table 4, the four structural equations of the system can be considered exactly identified. To estimate the structural parameters, we could use the two-stage least

squares method (2SLS), which solves the potential endogeneity problem (Gujarati and Porter 2008).

**Table 4.** Identification by order of the simultaneous equations model.

| Equation Number | K − k | m − 1 | K − k ≥ m − 1 | Identification |
|---|---|---|---|---|
| (1) | 9 − 8 | 1 | 1 ≥ 1 | Exactly identified |
| (2) | 9 − 8 | 1 | 1 ≥ 1 | Exactly identified |
| (3) | 9 − 8 | 1 | 1 ≥ 1 | Exactly identified |
| (4) | 9 − 8 | 1 | 1 ≥ 1 | Exactly identified |

Source: Authors' elaboration.

However, according to Henningsen and Hamann (2007), although the estimators obtained by the 2SLS method are consistent, the estimation by the three-stage least squares method (3SLS) presents estimators asymptotically more efficient, so we will use this method to obtain the structural parameters. Furthermore, the best estimation efficiency by the 3SLS method is obtained using the matrix of estimated moments of least squares of two stages of the structural disturbances to estimate the coefficients of the entire system simultaneously (Henningsen and Hamann 2007). Results are presented in Table 5 and will be discussed in the next section.

**Table 5.** Three-stage least squares regression.

| Equation | Obs | Parms | RMSE | "R-sq" | Chi | *p*-Value |
|---|---|---|---|---|---|---|
| LnWGH/LnWGM | 396 | 7 | 0.0252 | 0.9831 | 21.99 | 0.0012 |
| LnWDM/LnWGL | 396 | 7 | 0.0156 | 0.9747 | 91.8 | 0 |
| LnWGWM/LnWGWL | 333 | 7 | 0.0175 | 0.9712 | 37.96 | 0 |
| LnWGWM/LnWGWL | 333 | 7 | 0.0018 | 0.9618 | 89.35 | 0 |
| LnWGH/LnWGM | | Coefficient | | LnWGWH/LnWGWM | | Coefficient |
| LnWGM/LnWGL | | 0.38328 *** | | LnWGWM/LnWGWL | | 0.46209 *** |
| LnSBTC | | 0.06698 ** | | LnSBTC | | 0.06931 *** |
| LnUnion | | −0.10328 | | LnUnion | | 0.00639 |
| LnEPI | | −0.08973 | | LnEPI | | 0.00748 |
| LnEduc.Expend. | | 0.12328 *** | | LnEduc.Expend. | | 0.13951 ** |
| $LnCO_2$ | | 0.02257 ** | | $LnCO_2$ | | 0.02485 * |
| LnGDPpc | | −0.04477 ** | | LnGDPpc | | −0.05554 *** |
| Constant | | 1.22732 *** | | Constant | | 0.52147 *** |
| LnWGM/LnWGL | | | | LnWGWM/LnWGWL | | |
| LnSBTC | | 0.03711 *** | | LnSBTC | | 0.0277 *** |
| LnUnion | | −0.08281 * | | LnUnion | | −0.06781 * |
| LnEPI | | −0.01253 | | LnEPI | | 0.01643 * |
| LnEduc.Expend. | | 0.09327 *** | | LnEduc.Expend. | | 0.10947 *** |
| $LnCO_2$ | | 0.01725 ** | | $LnCO_2$ | | 0.01638 |
| LnKOF | | 0.02145 | | LnKOF | | 0.43712 |
| LnGDPpc | | −0.01998 ** | | LnGDPpc | | −0.02215 ** |
| Constant | | 0.93281 *** | | Constant | | 1.64690 *** |

Note: ***, **, and * denote statistical significance at the 1%, 5%, and 10% levels of significance, respectively. Source: Authors' calculations.

## 5. Discussion of the Results

As we can see in Table 5, there is a positive and significant influence of the wage difference between workers with complete secondary education concerning workers with qualifications lower than secondary education on the wage difference between workers with higher education and those with only moderate degrees. This explanation may be because workers with medium qualifications, through professional experience or training, can achieve higher wage rates due to the constant search by companies for higher qualifications, but this does not replace the qualifications obtained through tertiary education. This

reality is true for most workers in OECD countries and female workers. For the majority of workers, under the condition ceteris paribus, for every 1% increase in the wage gap between the averagely qualified in relation to the least qualified, the wage gap between the most qualified in relation to those qualified with secondary education increases by 0.38%, and in the case of women 0.46%. These results show a wage approximation between genders.

In all models, the proxy variable for the SBTC has a positive sign. It is still considered statistically significant, although its coefficients are not very high, as already found by Kristal and Cohen (2017). Despite the reduced impact, it is greater for the wage gap for more skilled workers than for the wage gap for less skilled workers. Regarding the impact of the SBTC on the majority of male and female workers, the impact is similar, although we can consider that the SBTC approach is bringing the wage gaps between the majority of workers and women closer together. In the absence of studies that relate the wage difference between the average of most workers and the average of women and many fewer that relate this reality considering three types of academic qualifications and from the point of view of the SBTC approach, we want to highlight the works of Rendall (2017) and Cerina et al. (2021). Both address the wage gap between men and women based on technological change and the SBTC approach. Rendall (2017) mentioned that the SBTC and the increase in women's education could explain the increase in women's participation in the labor market and explain more than half of the gender pay gap. On the other hand, Cerina et al. (2021) state that although, on average, women still face a wage disadvantage compared to men, technological changes and the increase in women's education have made it possible to reduce wage inequalities in terms of gender.

In the case of the wage gap between all more qualified workers and the wage gap of the same type of workers, but for females, the union influence does not play any significant role in setting wage rates. Regarding the wage gap of less qualified workers, there exists significance with negative effects. This situation may suggest that unions, through their demands, only influence the wage rates of less qualified workers, as verified by Western and Rosenfeld (2011). Concerning the EPI variable, this is not significant in three of the four estimated models, contributing only in a residual way to the increase in the salary gap in the case of women who have medium-level qualifications. Although there is previous evidence of a positive relationship between the EPI index and economic growth, this environmental indicator does not practically influence the wage gaps under study.

The variable spending on education, as verified in several studies (Muinelo-Gallo and Roca-Sagalés 2011; Antonczyk et al. 2018; Nogueira and Afonso 2018; Jacobs and Thuemmel 2022), is of great importance, being considered the one that most contributed to the increase in wage gaps in the four estimated models. In the case of all more qualified workers, as well as for female workers, it is estimated that ceteris paribus, a 1% increase in expenditure on education, the remuneration of skilled workers and those with higher education in relation to qualified with secondary education, increase by around 0.12% and almost 0.14%, respectively. Regarding the wage gap for all less skilled workers and women, it is estimated under the same conditions that wage rates will rise by around 0.09% and almost 0.11%, respectively. Despite being a slow movement, there is an approximation of the average wage rates of women concerning all workers, and Moore (2018) also reached these conclusions.

$CO_2$ emissions per capita have a positive impact, but only with a reduced coefficient in the formation of wage gaps, in three of the four estimated models. Only in the model that regresses the salary gap between women who have completed secondary education with respect to those who do not have this level of education is it not statistically significant. These results do not confirm what Krueger et al. (2021) advocated, who state that the more qualified workers are, the more they are willing to give up part of their wages due to environmental concerns.

Finally, in terms of GDP per capita, increases in average domestic income benefit less qualified workers compared to more qualified workers, bringing their wage rates closer together. Similar conclusions have already been reached by several authors, such

as Kuznets (1955), Clark et al. (2006), and Nogueira and Afonso (2018). In our case, this evidence is more noticeable among workers with higher education than among those with secondary education, than among the rest. In terms of female workers, the impact of this variable is very similar.

The limitations of this study are related to the time-space available in terms of data. Despite our sample covering 13 years, a possible higher existence of data could modify the conclusions or the intensity of the coefficients. Furthermore, as we said earlier, we can consider as another limitation that no previous literature relates the salary difference between the average of most workers and the average of women, considering three types of academic qualifications and from the point of view of the SBTC approach. A final limitation may be that the OECD countries present differences, which will only be visible when conducting individual analyses.

## 6. Conclusions and Policy Recommendations

Investments and increases in expenditure on R&D are generally accepted as driving forces for innovation, bringing with them reinforcements in competitiveness, productivity, economic growth, and wage increases. In this paper, we seek to address the formation of gaps in wage rates for workers in OECD countries, considering the SBTC approach, which is widely established in the economic literature related to the labor market.

This extensive empirical study spans 2007 to 2020 for OECD countries. It uses the wage gap between university and high school graduates and between high school graduates and below high school graduates. Similarly, we regress these same two wage rates for the case of female workers as a percentage of men's earnings. In addition to the variable that normally serves as a proxy for the SBTC, we used a set of control variables in the four regressions performed.

Since we suspect that there may be economic reasons for the existence of interdependence between the two wage gaps, we used simultaneous equation modeling, which choice proved to be right. One important conclusion we can draw, which is only possible using this estimation method, is that increases in the wage gap between high school graduates and below high school graduates will cause an increase in the wage gap between high school graduates and graduates of tertiary education. This fact is even more pronounced in female workers, causing a salary approximation between genders. This evidence proves that in terms of salary rates, tertiary education turns out to be more rewarding than experience and professional training. We can call this fact the "contagion effect" in wage rates, to the benefit of more qualified workers, serving as an incentive for workers to seek better qualifications.

Regarding the effect caused by the SBTC in formulating wage inequalities, as already mentioned, the result aligns with previous conclusions (Nogueira and Afonso 2018; Hutter and Weber 2022) and others. Increasingly qualified activities require workers with additional skills, and, as a result, their wage rate increases compared to those with lower qualifications. Therefore, acquiring additional skills continues to offset and widen wage gaps.

The results of union influence produce more effects on wage rates for workers with lower qualifications, reducing their wage gap with respect to those with high school graduates. This reality occurs with similar magnitude both in the case of the generality of workers and in the case of female workers. By opposition, with little influence in exacerbating or shortening relative wage rates, is the Environmental Performance Index, thus denoting its little importance of environmental issues for wage setting in the labor market. Still, concerning environmental issues and contrary to expectations, per capita $CO_2$ emissions contribute (albeit in a residual way) to the exacerbation of wage gaps. But to comply with climate agreements, countries must reduce their CO emissions, and the importance of this result becomes less and less impactful on the wage gap.

Once again, the expenditure made by countries on education proves to be the most important variable in promoting the increase in wage inequalities among workers, taking

into account their level of qualifications. Investments in education continue to promote wage improvements for more skilled workers. This leads to the conclusion that this type of investment should be increased to improve the living conditions of workers seeking higher qualifications. Additionally, it also turns out to be an invitation to students to continue and complete their university-level studies.

An important fact that has already been verified in several previous studies, the GDP per capita variable contributes to an approximation of wage rates among all workers. Increases in average earnings allow access to higher qualifications and seen in isolation via supply and demand mechanisms in the labor market, reduces wages as the supply of qualified work increases.

Regarding wage inequalities of female workers in relation to the majority of workers, the SBTC variable promotes, albeit in a small way, wage equality. More notably, expenditure on education has a more substantial impact on the approximation of the average wage gap for women in relation to the average for most workers. In the same sense, economic growth promotes the reduction of wage inequalities between female workers and the majority of workers.

Expenditure on education and economic growth promote two important facts: they contribute to the increase in the wage gap among workers, given their level of qualifications, inviting students to pursue their studies, and they also contribute in a significant way to reducing gender pay inequalities. In this way, countries must continue to encourage their students to acquire more and more skills, which in the first place, promotes increased productivity and, consequently, wage levels, as already verified, for example, by Nogueira and Afonso (2018). As spending on education contributes to reducing wage gaps between the majority of workers and women, initiatives that promote increased schooling should be reinforced within the scope of OECD countries, but more significantly in countries where this wage gap is more evident. A country-level study would allow us to highlight these differences and help delineate policy directions more specifically for each context. In the future, this work could be extended to other realities to do a comparative analysis since one of the constraints of the present work is to consider the OECD group solely. Different realities in terms of development and educative realities would benefit in terms of learning about wage gaps, especially considering inequalities among gender.

**Author Contributions:** Conceptualization, M.C.N. and M.M.; investigation, M.C.N.; methodology, M.C.N.; supervision, M.M.; writing—original draft, M.C.N.; writing—review and editing, M.M. All authors have read and agreed to the published version of the manuscript.

**Funding:** This research received no external funding.

**Institutional Review Board Statement:** Not applicable.

**Informed Consent Statement:** Not applicable.

**Data Availability Statement:** Data sharing is not applicable.

**Conflicts of Interest:** The authors declare no conflict of interest.

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
