# Peer review of "Skill-Biased Technological Change and Gender Inequality across OECD Countries—A Simultaneous Approach"

_economies, doi:10.3390/economies11040115_

Round 1

Reviewer 1 Report

Skill-biased Technological Change and gender inequality across OECD countries – a simultaneous approach

 Comments and Suggestions for Editor and Authors

This is an article that studies the wage gap between more and less skilled workers. It is a very relevant topic, in terms of research and contribution to knowledge. It is very important that the academic community develop research on the wage gap between more qualified and less qualified professionals, as well as its evolution over time, and on approaches to wage differences at the gender level (as is the case of this paper). The abstract, introduction, and literature review are carefully written and balanced. The methodology used and the discussion of the results are presented in a clear and objective way. The work is carried out and presented in a balanced way. The conclusion is presented objectively and moderately, without being long and tedious. The bibliography is extensive and current, with quality articles, therefore being of a high level. It is a good quality work overall and could be published, taking care of the little details for editing.

Precautions to be taken:

- check the text so that the tables are complete on a single page, to make them easier to read;

- there are bibliographical references that do not present the doi, and have it; example: Acemoglu, Daron, and David Autor. 2011.; Michaels, Guy, Ashwini Natraj and John Van Reenen. 2014. (among others).

Author Response

Answer:
We thank the reviewer for the appreciation made of our work. In accordance, we have ensured that tables are on a single page in the revised version and added the doi to all references that still do not have it. Thanks for the valuable suggestions.

Reviewer 2 Report

Thank you for the opportunity to review the article.

First of all, the study objective should be in the abstract section.

First sentence in paragraph 9: "To the best of our knowledge, ... ... ... paper, should be supported by evidence of reference collection in various academic databases. Also, why is this study important?

In the discussion section there should be an explanation regarding the comparison of the results with the results of similar studies that have been conducted previously on skill-based technological change and gender inequality.

What are the limitations of this study?

Good luck!

Author Response

Answer:
We would take the opportunity to say thanks to the reviewer for all the comments and valuable suggestions. Regarding this first query, we have rewritten the abstract in accordance. We advise for its new reading. Changes are marked in blue.
First sentence in paragraph 9: "To the best of our knowledge, ... ... ... paper, should be supported by evidence of reference collection in various academic databases. Also, why is this study important?
Answer:

We thank the reviewer for the valuable suggestion. This expression follows from the literature review provided where it was possible to confirm the novelty of the study. This one is stated clearly in the introduction in that same paragraph. Even so, we have added some more recent references, and if the reviewer has some more suggestions we would highly appreciated it.
In the discussion section there should be an explanation regarding the comparison of the results with the results of similar studies that have been conducted previously on skill-based technological change and gender inequality.
Answer:
Thank you for the valuable suggestion. We proceeded following the query and presented other authors' findings that reached similar conclusions.
What are the limitations of this study?
Answer:
Thanks for the suggestion. We have inserted within the conclusions the main limitations of the present study.
Good luck! Thanks!

Reviewer 3 Report

In Section 1: present the main objective of the research clearly and directly.

In Section 2, review the structure of the text: include subtopics based on keywords to facilitate reader understanding.

In Table 1, column "Source": include scientific articles.

Review the search title, based on the main objective and problem situation.

Author Response

Comments and Suggestions for Authors
In Section 1: present the main objective of the research clearly and directly.
Answer:
The research goal has been presented clearly as also suggested by another reviewer in the abstract directly to become clearer.
In Section 2, review the structure of the text: include subtopics based on keywords to facilitate reader understanding.
Answer:
Thanks for the suggestion. We tried to proceed in accordance and divide by subtopics.
In Table 1, column "Source": include scientific articles.
Answer:
Thanks for the suggestion. The references have been added.

Review the search title, based on the main objective and problem situation
Answer:
We thank the reviewer for the suggestion. We believe the title fits the goals of the present article and the conclusions reached.

Reviewer 4 Report

This paper aims to explain the constant increase in wage inequalities between more and less skilled workers, including female workers. In addition, it incorporates a wide range of control variables, estimating the wage gap through four models.

In the introduction section, there are an excessive number of citations to the same groups of authors. I would suggest the inclusion of recent publications, from a wide variety of authors. 

The literature review is well presented, citing references within the last 5 years, altogether with an expressive number of others older studies. However, I do not find any discussion about using the structural equation in this subject. I suggest the inclusion of this analysis in the review.  

Section 3, Data, Variables, Statistics, and Correlations are shown in an appropriate format, in an easy way to interpret. Tables 1, 2, 3, and 4 present the source, but the source of table 5 is missing. 

In section 4, Empirical Analysis. Model specification and Estimation Methods, the authors explain their idea of using a “the econometric estimates were therefore carried out using a system of simultaneous equations, with the structural form of the equations” (lines 438-439). Also, they affirm: “The approach through simultaneous equations makes it possible to analyze not only 9 the individual behavior of each equation but also the possible relationships between 10 equations and variables, as well as the dynamic relationships that occur between them, in 11 a given period. (Lines 9-11, page 15). Through that, I would like to suggest authors address the “dynamic relationships” in the results (Section 5). The results are presented but I did not identify the dynamic relationship as it was specifically mentioned here. 

In section 6, Conclusion and Policy Recommendation, I found conclusions consistent with the evidence, but the “policy recommendation” is not clear (at least for me), and it could be better delineated, providing arguments based not only on the results but also on the literature review done previously. Authors may cover topics such as: What this study could contribute in terms of policies for countries (or specific groups of countries) among all those in the OCDE? Are there differences in those countries that would imply fragilities in this study? What are the limitations of the findings? 

Author Response

Comments and Suggestions for Authors
This paper aims to explain the constant increase in wage inequalities between more and less skilled workers, including female workers. In addition, it incorporates a wide range of control variables, estimating the wage gap through four models.
In the introduction section, there are an excessive number of citations to the same groups of authors. I would suggest the inclusion of recent publications, from a wide variety of authors.
Answer:
We would like to say thanks to the reviewer for the valuable suggestion. We have proceeded in accordance. In the introduction, we have inserted more citations.

The literature review is well presented, citing references within the last 5 years, altogether with an expressive number of others older studies. However, I do not find any discussion about using the structural equation in this subject. I suggest the inclusion of this analysis in the review.
Answer:
Thanks for the valuable comments. We have used simultaneous and non-structural ones and this constitutes another of the innovative contributions of the article. For this reason, literature is not included due to its lack of existence within the specific subject.
Section 3, Data, Variables, Statistics, and Correlations are shown in an appropriate format, in an easy way to interpret. Tables 1, 2, 3, and 4 present the source, but the source of table 5 is missing.
Answer:
Thanks for calling our attention to this fact. We have added the source now in table 5.
In section 4, Empirical Analysis. Model specification and Estimation Methods, the authors explain their idea of using a “the econometric estimates were therefore carried out using a system of simultaneous equations, with the structural form of the equations” (lines 438-439). Also, they affirm: “The approach through simultaneous equations makes it possible to analyze not only the individual behavior of each equation but also the possible relationships between equations and variables, as well as the dynamic relationships that occur between them, in a given period. (Lines 9-11, page 15). Through that, I would like to suggest authors address the “dynamic relationships” in the results (Section 5). The results are presented but I did not identify the dynamic relationship as it was specifically mentioned here.
Answer:
Thanks for the comment. It's already fixed. There was an oversight on our part, as they are not dynamic relationships and this has been corrected in accordance.
In section 6, Conclusion and Policy Recommendation, I found conclusions consistent with the evidence, but the “policy recommendation” is not clear (at least for me), and it could be better delineated, providing arguments based not only on the results but also on the literature review done previously. Authors may cover topics such as: What this study could contribute in terms of policies for countries (or specific groups of countries) among all those in the OCDE? Are there differences in those countries that would imply fragilities in this study? What are the limitations of the findings?

Answer:
Thanks for the valuable comments and suggestions. We have proceeded in accordance in the results section by discussing policy directions, and we have now inserted limitations since it was also requested by another reviewer, where all these queries have been addressed.

Round 2

Reviewer 1 Report

Dear Editor and Authors,

After having read the latest version of the paper "Skill-biased Technological Change and gender inequality across OECD countries - a simultaneous approach", I have nothing else to suggest. The paper is well-written and should be published by the journal Economies.

Best regards.

Author Response

Dear Reviewer,

Thanks for reviewing the paper again. 

Reviewer 2 Report

Dear authors,

As has been added in the discussion section, namely there are two references (Cerina et al, 2021 and Renall, 2017), there should be an explanation of the relevance of them to the study being presented by the authors in the literature review section.

Are the two references consistent with the new explanation of the limitations of the research described, 'Another limitation ... ... ... no previous literature ... .... ... ... approach?

Good luck!

Author Response

Reviewer answer for:

[Economies] Manuscript ID: economies-2230607 - Major Revisions

Manuscript ID: economies-2230607

Title: Skill-biased Technological Change and gender inequality across OECD countries – a simultaneous approach

Received: 4 February 2023

Dear authors,

As has been added in the discussion section, namely there are two references (Cerina et al, 2021 and Renall, 2017), there should be an explanation of the relevance of them to the study being presented by the authors in the literature review section.

Are the two references consistent with the new explanation of the limitations of the research described, 'Another limitation ... ... ... no previous literature ... .... ... ... approach?

Good luck!

Dear Reviewer,

Thanks for reviewing the paper again. The changes we made, following your suggestions, can be found in the paper in blue.

We refer to the importance of including the two works you have mentioned and we improve the limitations in this regard as well.